# Ex Uno Pluria: Insights on Ensembling in Low Precision Number Systems

**Giung Nam**
Kim Jaechul Graduate School of AI
KAIST, Daejeon, South Korea
`giung@kaist.ac.kr`

**Juho Lee**
Kim Jaechul Graduate School of AI
KAIST, Daejeon, South Korea
`juholee@kaist.ac.kr`

## Abstract

While ensembling deep neural networks has shown promise in improving generalization performance, scaling current ensemble methods for large models remains challenging. Given that recent progress in deep learning is largely driven by the scale, exemplified by the widespread adoption of large-scale neural network architectures, scalability emerges an increasingly critical issue for machine learning algorithms in the era of large-scale models. In this work, we first showcase the potential of low precision ensembling, where ensemble members are derived from a single model within low precision number systems in a training-free manner. Our empirical analysis demonstrates the effectiveness of our proposed low precision ensembling method compared to existing ensemble approaches.

## 1 Introduction

In computer science, it is a de facto standard to represent continuous real numbers using finite precision number systems. While many applications rely on precision formats like FP32 or FP64, the deep learning community is increasingly turning to 16-bit floating-point formats such as FP16 (Micikevicius et al., 2018) or BF16 (Dean et al., 2012) to reduce memory usage during training. More recently, researchers are further exploring low precision optimization, aiming to utilize fewer bits (8 bits or less) to represent weights, activations, and gradients throughout the training process (Gupta et al., 2015; Li et al., 2017; Sun et al., 2020; Wortsman et al., 2023).

While low precision number systems can aid in training deep neural networks, they are also beneficial for reducing inference costs in real-world deployments of such models (Jacob et al., 2018). In particular, recent advancements in large language models (Brown et al., 2020; Touvron et al., 2023) containing billions of parameters have triggered active exploration of *post-training quantization* techniques, for deploying pre-trained large language models on hardware with limited memory resources. This exploration encompasses quantizing both weights and activations (Dettmers et al., 2022; Yao et al., 2022), as well as quantizing weights only (Frantar et al., 2023; Lin et al., 2024).

Originally, the goal of post-training quantization is to specify the best solution represented in low precision number systems, aiming to reduce discrepancies from the original high-precision weights, like perturbations in weight values or increases in loss functions (Nagel et al., 2020). On the other hand, the presence of numerous distinct yet high-performing models within a single basin on loss landscapes (Sadrtdinov et al., 2023; Lion et al., 2024) evokes a Bayesian concept of *marginalization instead of optimization*, which involves utilizing multiple solutions rather than relying solely on one solution (Wilson and Izmailov, 2020).

Hinging on this insight, we suggest building ensembles within low precision number systems, as illustrated in Fig. 1. It depicts a proof-of-concept method for ensemble construction using stochastic rounding, a technique commonly used in low precision training to address the issue of rounding

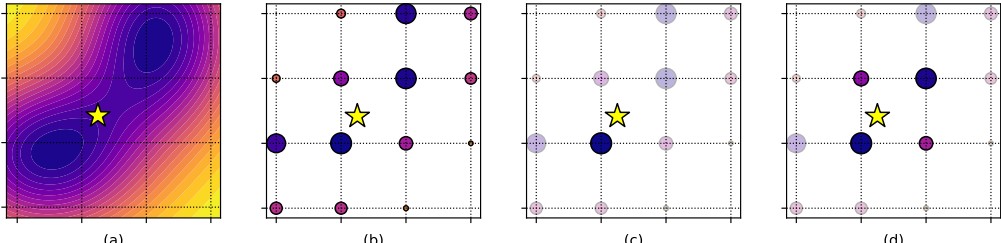

**Figure 1: Concepts of low precision ensembling.** It shows a two-dimensional schematic, where the x and y axes represent the neural network weights, while the contours above visualize the loss surface. (a) Let the pre-trained weights, denoted by a yellow star-shaped marker (☆), be positioned within a basin on the loss landscape. In general, (b) post-training quantization methods introduce lower precision number systems, and then (c) choose one candidate from the system, such as the nearest one. (d) However, there are many other highly effective models available, that can contribute to ensemble predictions.

weight updates below the minimum precision threshold to zero (Gupta et al., 2015). In our approach, stochastic rounding is employed for *ensembling* rather than optimization. Indeed, our experimental findings in Section 4 validate that low precision ensembling improves the downstream performance of pre-trained large models without any further training on downstream data.

Confirming the potential of low precision ensembling for pre-trained models, we extend our investigation through a comparative study with existing methods involving ensembling. Specifically, we examine Bayesian approaches that approximate a Gaussian posterior over the loss landscape (Maddox et al., 2019; Shen et al., 2024), and sampling techniques that collect model copies from the update trajectory (Huang et al., 2017; Zhang et al., 2020). Our experimental results in Section 4 show that low precision ensembling successfully gathers diverse ensemble members contributing the final ensemble performance within the low precision number system.

The main contributions of our work can be outlined as follows:

- Introducing low precision number systems inevitably results in quantization errors, usually seen as a flaw to be corrected in neural network quantization. Our work presents a novel viewpoint: these errors can be utilized as a source to improve ensemble diversity. Expanding on our comprehension of diversity, we suggest a simple yet powerful approach to constructing ensembles called Low Precision Ensembling with Bernoulli Stochastic Rounding (LPE-BSR), particularly advantageous for large models.

- The proposed LPE-BSR involves assembling ensemble members with low precision number systems, effectively addressing a notable challenge associated with memory costs inherent in ensemble methods. In this regard, our work holds promise for utilizing low precision number systems to construct ensembles of large models, offering a potential solution for the scalability issue faced by the Bayesian deep learning community in the era of large-scale models (Papamarkou et al., 2024).

## 2 Ensemble methods in modern transfer learning scenarios

Ensembling neural networks is a long-established idea in machine learning (Hansen and Salamon, 1990), where the underlying design principle is to create a strong hypothesis that effectively explains the data by combining a set of weak hypotheses (Kearns, 1988). Notably, even after the transition into the deep learning era, ensemble methods continue to serve as a straightforward yet powerful strategy for boosting the performance of machine learning algorithms involving deep neural networks (Krizhevsky et al., 2012; Ciresan et al., 2012). However, the operational principles of such *deep ensembles*, i.e., ensembles composed of deep neural networks, deviate from those of classical statistical models and remain not fully comprehended. For instance, both Lee et al. (2015) and Nixon et al. (2020) validated that *bagging* (Breiman, 1996), built on the theoretically well-motivated *bootstrap* method (Efron, 1992), does not offer any benefits over the simplest deep ensembles consisting of models obtained from multiple training runs with different random seeds.

Empirically, it is well-known that we need nothing more than employing different initializations for each ensemble member to construct high-performance deep ensembles (Lakshminarayanan et al., 2017). Fort et al. (2019) delved deeper into this and emphasized the significant role played by a highly non-convex loss function of deep neural networks, where varying initializations for stochastic optimization yield different functional modes. It also aligns with the Bayesian perspective provided by Wilson and Izmailov (2020), which suggests that deep ensembles are involved in approximating multimodal posterior distribution to the Bayesian model averaging. To sum up, while the operational principle of deep ensembles may differ from classical ensembles, the underlying idea regarding *diversity* remains constant (Krogh and Vedelsby, 1994; Ortega et al., 2022), i.e., ensembles demonstrate improved performance when their individual members offer *diverse* predictions.

However, the simple strategy mentioned earlier, which aims to cover multiple modes on the loss landscape by starting from different initializations to achieve ensemble diversity (Fort et al., 2019; Wilson and Izmailov, 2020), faces challenges in modern transfer learning scenarios. It arises from the typical situation that there is only *one* pre-trained model available for fine-tuning; due to the considerable cost for pre-training of large models, model providers usually do not distribute multiple model copies. In such cases, fine-tuned solutions originating from the same pre-trained weights often inhabit the same pre-trained basin, leading to restricted exploration within the loss landscape (Neyshabur et al., 2020; Mustafa et al., 2020). Consequently, our attention should be directed towards addressing the *local* structure around the pre-trained basin in modern transfer learning scenarios (Wortsman et al., 2022; Sadrtdinov et al., 2023; Lee et al., 2024), rather than the *global* multimodal structure of the loss landscape (Fort et al., 2019; Wilson and Izmailov, 2020).

## 3 Preliminaries

**Finite precision number systems.** Computers use binary sequences for data encoding, with FP32 serving as the primary finite precision number system employed to represent real numbers. Given its coverage from approximately $10^{-38}$ to $10^{38}$ with a resolution of $10^{-7}$, we consider the FP32 system as the continuous set of real numbers $\mathbb{R}$. Moreover, we describe INT-$B$ systems, commoly utilized low precision number systems for neural network quantization:

$$\mathbb{F}_{\text{INT-}B} = \left\{ m \times \frac{w_{\text{absmax}}}{2^{B-1} - 1} \ : \ m \in \left\{ -2^{B-1} + 1, \dots, 0, \dots, 2^{B-1} - 1 \right\} \right\}, \tag{1}$$

where the system can represent real numbers in the range $[-w_{\text{absmax}}, w_{\text{absmax}}]$ with a resolution of $w_{\text{absmax}}/(2^{B-1} - 1)$, and the integer $m$ can be encoded using $B$ bits. This is the simplest form of integer quantization, with possible variations like a zero offset or non-uniform resolution (Gholami et al., 2021; Yvinec et al., 2023). Unless otherwise specified, our experimental results employ this basic form of *symmetric uniform quantization* with $B = 5$ for simplicity. We also use *per-channel granularity*, sharing the number systems among the output channels of each linear layer.

**Rounding rules.** Let $\mathbb{F} \subset \mathbb{R}$ be a finite precision number system. For any $w \in \mathbb{R}$, there are two rounding options in practice: $\lfloor w \rfloor = \max \{ \hat{w} \in \mathbb{F} : \hat{w} \leq w \}$ and $\lceil w \rceil = \min \{ \hat{w} \in \mathbb{F} : \hat{w} \geq w \}$. The *rounding-to-nearest* (RTN) scheme deterministically selects the closest one, i.e., $\hat{w} = \lfloor w \rceil$, whereas the *Bernoulli stochastic rounding* (BSR) scheme randomly chooses one of them, i.e.,

$$\hat{w} = \lfloor w \rfloor \text{ with probability } \lceil w \rceil - w, \text{ or } \hat{w} = \lceil w \rceil \text{ with probability } w - \lfloor w \rfloor. \tag{2}$$

Observing empirically that BSR sometimes produce superior results compared to RTN motivates the neural network quantization community to explore more sophisticated rounding schemes (Nagel et al., 2020; Lee et al., 2023). On the other hand, recognizing the presence of *multiple* competitive solutions, we consider leveraging them for low precision ensembling.

**Ensemble methods.** In ensemble methods for classification problems, the final prediction during testing is obtained by combining $S$ predictions:

$$p(y|x) = \frac{1}{S} \sum_{s=1}^{S} p(y|x, \boldsymbol{w}_s), \tag{3}$$

where $y$ is a class label, $x$ is an input, and $p(y|x, \boldsymbol{w}_s)$ is the categorical probabilities predicted by the $s^{\text{th}}$ member, which is a neural network model with parameters $\boldsymbol{w}_s$. These $\boldsymbol{w}_s$ could either be *maximum a posteriori* (MAP) solutions obtained through multiple stochastic optimizations (Lakshminarayanan et al., 2017), or they might be intermediate checkpoints from a single training run (Huang et al., 2017; Garipov et al., 2018).

**Table 1: Motivating results for low precision ensembling of pre-trained ViT models.** Negative log-likelihood (NLL), classification error (ERR), and ensemble ambiguity (AMB) for rounding-to-nearest (RTN) and low precision ensembling with Bernoulli stochastic rounding (LPE-BSR) derived from the publicly available pre-trained ImageNet model (☆). Blue highlights the areas where LPE-BSR excels, particularly in larger models and lower precision settings.

| Method | System | ViT-T/16 (6M) | | | ViT-S/16 (22M) | | | ViT-B/16 (87M) | | | ViT-L/16 (307M) | | |
|---|---|---|---|---|---|---|---|---|---|---|---|---|---|
| | | NLL | ERR | AMB | NLL | ERR | AMB | NLL | ERR | AMB | NLL | ERR | AMB |
| ☆ Pre-trained | FP32 | .932 | .243 | - | .667 | .185 | - | .687 | .182 | - | .639 | .165 | - |
| RTN | INT-6 | .948 | .247 | - | .671 | .185 | - | .687 | .182 | - | .639 | .165 | - |
| | INT-4 | 1.23 | .315 | - | .822 | .218 | - | .716 | .184 | - | .647 | .167 | - |
| LPE-BSR | INT-6 | .932 | .245 | .024 | .665 | .185 | .014 | .681 | .181 | .006 | .632 | .164 | .003 |
| | INT-4 | 1.30 | .298 | .489 | .821 | .211 | .268 | .648 | .175 | .088 | .600 | .160 | .037 |

In the Bayesian framework, neural network weights $w$ are treated as random variables, and $w_s$ are seen as Monte Carlo samples employed to approximate the posterior distribution. More precisely, Eq. 3 can be seen as a simple Monte Carlo method to approximate the posterior with a set of point masses, where the locations are given by samples from another approximate posterior $q$, i.e., $p(w|\mathcal{D}) \approx \sum_{s=1}^{S} \delta(w - w_s)/S$, $w_s \sim q(w)$, where $\mathcal{D}$ denotes the data and $\delta$ is the Dirac delta function (Wilson and Izmailov, 2020). A common practice is to use a Gaussian approximation $q(w) = \mathcal{N}(w; \mu, \Sigma)$ to generate samples $w_s \sim q(w)$ in a tractable manner (Maddox et al., 2019; Shen et al., 2024), and then approximate the predictive distribution by computing

$$p(y|x) = \frac{1}{S} \sum_{s=1}^{S} p(y|x, w_s), \quad w_1, \ldots, w_S \sim q(w). \tag{4}$$

## 4 An empirical study of low precision ensembling

We present a simple yet effective low precision ensemble construction strategy, Low Precision Ensembling with Bernoulli Stochastic Rounding (LPE-BSR), which computes Eq. 4 using

$$q(w) = \prod_{i=1}^{D} q(w^{(i)}), \quad q(w^{(i)}) = \lambda_i \cdot \delta(w^{(i)} - \lfloor w^{(i)} \rfloor) + (1 - \lambda_i) \cdot \delta(w^{(i)} - \lceil w^{(i)} \rceil), \tag{5}$$

for $i = 1, \ldots, D$. Here, $D$ denotes the number of neural network parameters, where each per-channel parameter group shares the same low precision number system, as explained in Section 3. Using the rounding operations $\lfloor \cdot \rfloor$ and $\lceil \cdot \rceil$ defined within each system, the probability is determined by $\lambda_i = \lceil w_i \rceil - w_i$, as in Eq. 2. Certainly, the proposed LPE-BSR is not Bayesian, and we have simply expressed it in the form of Eq. 4 for the sake of notational simplicity.

### 4.1 Motivation: training-free ensemble construction of large ViT models

We begin with motivating experiments using the publicly available series of pre-trained vision transformer models (ViT; Dosovitskiy et al., 2021). Detailed information about each model can be found in Appendix A. Table 1 summarizes the evaluation results on a subset of the ImageNet validation split, along with the parameter count for each model. In this context, the pre-trained model corresponds to the star-shaped marker (☆) in Fig. 1, with RTN using the nearest value in low precision number systems as shown in Fig. 1(c), and LPE-BSR forming an ensemble by selecting $S = 10$ nearby samples as illustrated in Fig. 1(d).

Table 1 provides the following key findings: 1) Larger models experience less performance degradation when reducing the precision of numerical systems. More precisely, in the RTN results, the classification error increases from .243 → .247 → .315 when transitioning from FP32 → INT-6 → INT-4 at ViT-T/16, whereas at ViT-L/16, it shifts from .165 → .165 → .167. 2) Lower precision systems introduce diversity among samples in LPE-BSR. Specifically, ensemble ambiguity is the metric for quantifying the ensemble diversity (to be defined in Section 4.3), and in all models, the ensemble ambiguity increases when transitioning from INT-6 → INT-4.

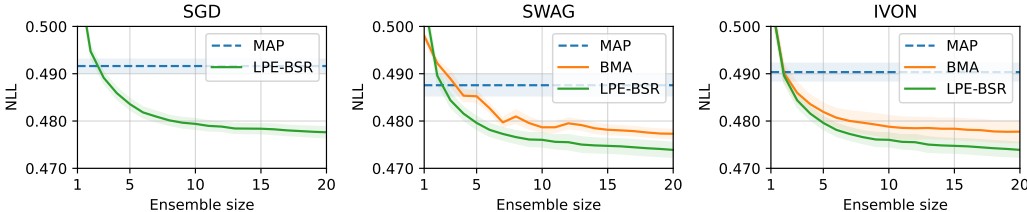

**Figure 2: Comparing low precision ensembling to Bayesian methods.** Negative log-likelihood for Bayesian model averaging using an approximate Gaussian posterior derived from SWAG or IVON (BMA, shown in orange) and low precision ensembling with Bernoulli stochastic rounding centered around the MAP solution obtained by each optimizer (LPE-BSR, shown in green).

Importantly, these findings are in line with the core principle of ensemble methods being effective: a key condition for an ensemble of classifiers to outperform any of its individual members is when the classifiers are both *accurate* and *diverse* (Dietterich, 2000). When introducing the low precision number system, 2) suggests that diversity can be achieved by leveraging the quantization error inherent in this process, while 1) emphasizes that larger models maintain accurate individual performance throughout this process. With this compelling motivation for low precision ensembling established, we now proceed to compare it with existing ensemble methods.

## 4.2 Comparative study to Bayesian methods

Our initial investigation into the proposed LPE-BSR aims to assess the effectiveness of collecting ensemble members within the discrete space defined by the low precision number system. While using fewer bits to represent samples would certainly reduce ensemble costs, there is a concern that we might overlook potentially good ensemble candidates outside this discrete space. To address this, we conduct a comparative study with two Bayesian deep learning methods: improved variational online newton (IVON; Shen et al., 2024) and stochastic weight averaging Gaussian (SWAG; Maddox et al., 2019). Both methods use samples drawn from an approximate Gaussian posterior in the continuous weight space and perform ensembling in a Bayesian manner.

To sum up our experiment, we first fine-tune the zero-shot CLIP-ViT model (Radford et al., 2021) on the ImageNet training split to obtain the MAP solution $w^*_{\text{MAP}}$. For LPE-BSR, this fine-tuning can employ any of the SGD, SWAG, or IVON optimizers. We then define $q$ according to Eq. 5 using $w^*_{\text{MAP}}$ and compute Eq. 4 with $S$ samples. For SWAG and IVON, $q(w) = \mathcal{N}(w; \mu, \Sigma)$ is defined using the $\mu = w^*_{\text{MAP}}$ and $\Sigma$ obtained during their respective optimization processes, followed by the computation of Eq. 4. Consequently, LPE-BSR samples neighboring points of $w^*_{\text{MAP}}$ within the discrete space defined by the low precision number system, whereas SWAG and IVON sample nearby points of $w_{\text{MAP}}$ in the continuous space with Gaussian noise added. For more details on SWAG and IVON, including their hyperparameters, please refer to Appendix E.

Fig. 2 presents the outcomes of our experiments conducted with CLIP-ViT-L/14. In our experimental setup, both SWAG and IVON successfully estimated both the MAP mean and the covariance matrix, resulting in a lower negative log-likelihood compared to MAP through Bayesian model averaging, as illustrated in the second and third subplots of Fig. 2. Remarkably, our LPE-BSR, derived from the MAP solution obtained by the SGD optimizer, achieves competitive results with Bayesian model averaging through SWAG or IVON. Moreover, when using the same MAP solution obtained with the SWAG or IVON optimizer for a fair comparison, it even outperforms these methods.

From a numerical integration perspective (Wilson and Izmailov, 2020; Wilson, 2021), the conditions for successful approximate Bayesian inference in deep learning are very similar to those for successful ensembling, as discussed in Section 4.1 with reference to Dietterich (2000). Specifically, it entails 1) finding *typical* points in the posterior that represent regions of substantial mass (cf. *accurate*); and (ii) ensuring a *diverse* set of points to give rise to different functions (cf. *diverse*). Consequently, we proceed to conduct a comparative analysis of these two factors concerning our LPE-BSR method and the Bayesian methods we considered, SWAG and IVON.

**Table 2: Results for low precision ensembling of fine-tuned models.** We compute (a) average loss, (b) ambiguity, and (c) ensemble loss for diversity analysis, along with evaluation metrics to assess overall performance. Both BMA and LPE-BSR samples are centered around ☆ within each group (MAP in this context), which are separated by horizontal lines.

| Optimizer | Method | System | Diversity analysis (a) | (b) | (c) | Evaluation metrics NLL | ERR | ECE |
|---|---|---|---|---|---|---|---|---|
| SWAG | ☆ MAP | FP32 | .488±.003 | - | .488±.003 | .488±.002 | .137±.001 | .034±.001 |
| | BMA | FP32 | .498±.002 | .015±.000 | .483±.001 | .477±.002 | **.136**±.001 | **.021**±.001 |
| | LPE-BSR | INT-6 | .492±.003 | .007±.000 | .485±.003 | .483±.002 | **.136**±.001 | .031±.001 |
| | | INT-5 | .507±.002 | .026±.000 | .481±.002 | **.473**±.002 | **.136**±.001 | **.021**±.001 |
| | | INT-4 | .643±.004 | .129±.001 | .514±.003 | .513±.002 | .146±.001 | .027±.000 |
| IVON | ☆ MAP | FP32 | .490±.002 | - | .490±.002 | .489±.001 | .136±.001 | .037±.001 |
| | BMA | FP32 | .503±.001 | .021±.000 | .483±.001 | .475±.001 | **.135**±.000 | .026±.000 |
| | LPE-BSR | INT-6 | .492±.001 | .006±.000 | .486±.001 | .483±.001 | **.135**±.000 | .033±.001 |
| | | INT-5 | .507±.001 | .025±.000 | .481±.001 | **.472**±.001 | **.135**±.000 | **.023**±.000 |
| | | INT-4 | .642±.003 | .131±.001 | .512±.002 | .509±.001 | .145±.001 | .026±.001 |
| SGD | ☆ MAP | FP32 | .492±.002 | - | .492±.002 | .492±.002 | .138±.000 | .035±.001 |
| | LPE-BSR | INT-6 | .495±.001 | .006±.000 | .488±.001 | .485±.001 | .138±.000 | .030±.000 |
| | LPE-BSR | INT-5 | .513±.001 | .057±.001 | .456±.000 | **.477**±.001 | **.137**±.000 | **.020**±.001 |
| | LPE-BSR | INT-4 | .663±.003 | .120±.001 | .544±.002 | .526±.001 | .150±.001 | .029±.001 |

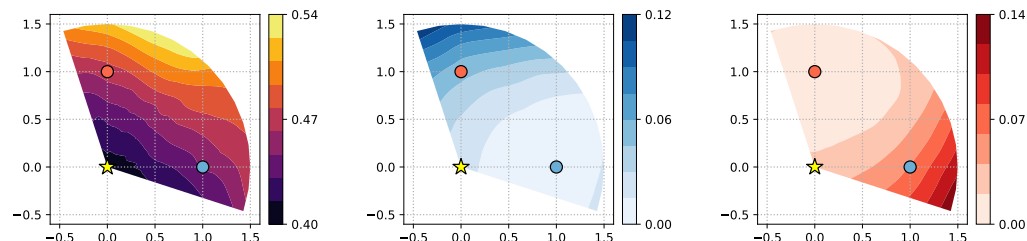

**Figure 3: Comparison between IVON and LPE-BSR samples.** Radial landscape plots visualize a plane subspace defined by three points: the MAP obtained by IVON (depicted as a yellow star ☆), samples in BMA and LPE-BSR procedures (represented by blue and red circle markers ○).

## 4.3 Diversity analysis of ensemble methods

Quantitative assessment of ensemble diversity is an essential metric for evaluating ensemble methods. To this end, we adopt the *generalized ambiguity decomposition* for cross-entropy loss presented by Wood et al. (2023), which can be easily measured using logit ensembling instead of probability ensembling. It should be noted that logit ensembling is solely utilized for diversity analysis, whereas probability ensembling is used for all other experimental results to ensure a fair comparison with Bayesian methods that compute the categorical predictions using the BMA integral. Please refer to Appendix B for more details on our diversity analysis.

Table 2 presents our experimental outcomes for the generalized ambiguity decomposition (cf. 'Diversity analysis'), along with the final ensemble perfomance (cf. 'Evaluation metrics'). Lowering the precision of numerical systems naturally amplifies quantization error, as evidenced by the results in the INT-$B$ rows, where reducing $B$ results in higher (a) average loss. However, this also coincides with an increase in ensemble diversity, with smaller $B$ values resulting in greater (b) ambiguity. Consequently, the superior (c) ensemble loss at an appropriate precision level (e.g., $B = 5$) highlights the fundamental principle of the low precision ensembling: *it does not merely perceive quantization error problematic, but rather utilizes it to obtain ensemble diversity*. Consequently, the proposed LPE-BSR yields improvements in evaluation metrics, as depicted in Table 2. Definitions for each metric can be found in Appendix B.

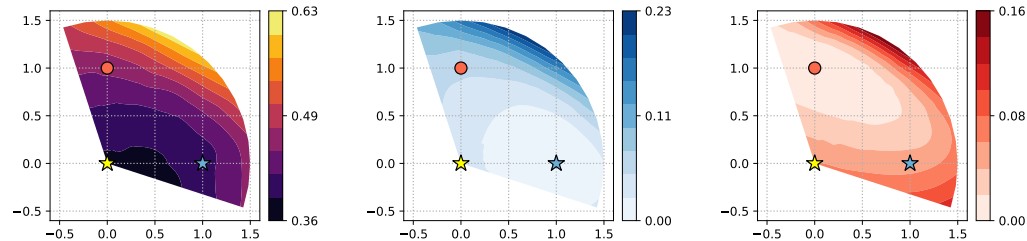

**Figure 4: Comparison between snapshot and LPE-BSR samples.** Radial landscape plots visualize a plane subspace defined by three points: the first and second snapshot samples obtained by SSE (represented by yellow and blue star-shaped marker ☆), and LPE-BSR sample derived from the first snapshot (depicted as a red circle ○).

Drawing inspiration from Fort et al. (2019), we further provide the radial landscape plot depicting a plane within weight space containing three points (shown as yellow, blue, and orange markers). The z-values for each subplot represent the negative log-likelihood (displayed on the left in a magma colormap), the function differences from the blue model (shown in the center in a blue colormap), and the red model (presented on the right in a red colormap). Namely, the first subplot indicates the placement of each model within the loss landscape, while the subsequent two subplots illustrate the extent to which they differ from each other.

Fig. 3 depicts radial landscape plots comparing IVON and LPE-BSR samples, showing their parallel roles in ensemble construction. Both show slightly higher individual negative log-likelihoods, shown by the circle markers, compared to the MAP denoted by a star-shaped marker in the first subplot, while also offering diverse function outputs as demonstrated in the subsequent subplots. Ultimately, LPE-BSR can identify samples in a low precision number system that qualitatively resemble the high-quality posterior samples provided by IVON, which leverages Hessian estimate information to compute the covariance of the approximate Gaussian posterior.

### 4.4 Combining with fast ensembling methods

A key advantage of LPE-BSR is its ability to gather ensemble members without requiring any backward computation. This feature, which eliminates the need for training, aligns with fast ensembling techniques aimed at enhancing the training efficiency of ensemble construction processes (Huang et al., 2017; Garipov et al., 2018; Benton et al., 2021). Consequently, we conduct empirical analysis to further investigate this alignment with snapshot ensembling (SSE; Huang et al., 2017), as well as cyclical stochastic gradient Langevin dynamics (CSGLD; Zhang et al., 2020), a closely related Bayesian posterior sampling algorithm. Both methods involve collecting snapshot samples on the loss landscape around $w_{\mathrm{MAP}}$ using a cyclical learning rate schedule. For more details, including hyperparameters, please refer to Appendix E.

We first verify whether LPE-BSR can generate an ensemble component distinct from SSE snapshots using radial landscape analysis. Fig. 4 illustrates a plane subspace containing the first and second SSE snapshots (displayed as star-shaped markers) and the LPE-BSR sample obtained from the first snapshot (represented by a circle marker). Indeed, LPE-BSR provided a novel sample that could contribute to the ensemble; it is clearly diverse from the existing snapshots, as shown in the second and third subplots, while achieving reasonably low individual negative log-likelihoods, as shown in the first subplot. By using such LPE-BSR samples along with SSE snapshots to build an ensemble, we can reduce the cost of achieving target performance in fast ensembling methods or attain better results with the same training budgets.

However, while fast ensembling techniques usually prioritize evaluating training budgets, particularly the number of backward passes as seen in the literature (Huang et al., 2017; Garipov et al., 2018; Zhang et al., 2020), we also consider memory budgets in our analysis—the total number of bits required to represent the entire ensemble model, which grows with the addition of ensemble members in fast ensembling. From this perspective, we devised a method to eliminate heavy SSE snapshots with high precision from the final ensemble; as a result, each original SSE snapshot is replaced by $S = 5$ LPE-BSR samples. This policy also aligns with our reserach objective of exploring

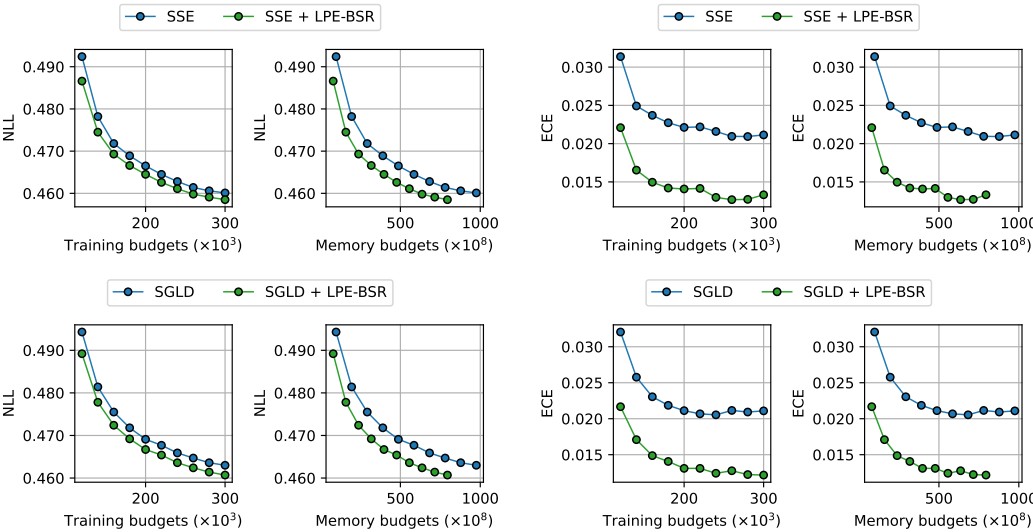

**Figure 5: Combining with fast ensembling methods.** Negative log-likelihood and expected calibration error for fast ensembling methods, SSE and CSGLD, in terms of training budgets, i.e., the number of backward passes, and memory budgets, i.e., the total number of bits for representing ensemble. **Top:** Results with SSE. **Bottom:** Results with CSGLD.

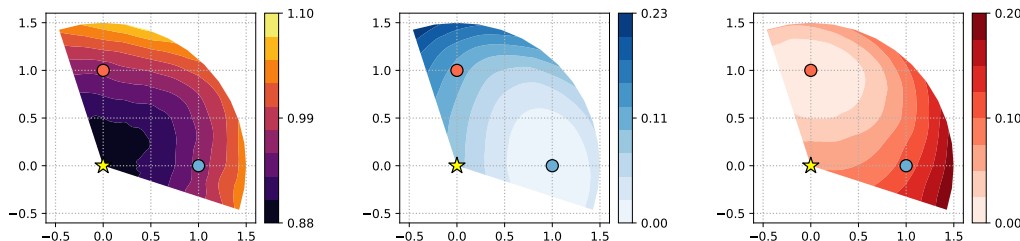

**Figure 6: Radial landscapes for zero-shot CLIP-ViT-L/14 model.** Radial landscape plots visuaslize a plane subspace defined by three points: a pre-trained model (depicted as a yellow star-shaped marker ☆) and two LPE-BSR samples derived from the pre-trained weights (represented by blue and red circle markers ○).

ensembling in low precision number systems. Consequently, by forming the ensemble exclusively with LPE-BSR samples in the low precision system, we attained better outcomes compared to SSE regarding both training budgets and memory budgets, as shown in Fig. 5.

## 4.5 Training-free ensemble construction of pre-trained large models

Our investigation so far, involving deep neural networks up to 300M in size, substantiates the efficacy of the proposed LPE-BSR methodology. In this section, we further extend our validation to confirm that LPE-BSR enables effective ensemble construction without training, even in larger models. To this end, in addition to 300M-scale CLIP-ViT-L/14 model (Radford et al., 2021), we employ a 1B-scale CLIP-ViT-G/14 model (Cherti et al., 2023) and an 8B-scale LLaMa model (Touvron et al., 2023) in a zero-shot manner. Appendix A provides public links for each model.

We first present a radial landscape analysis of the CLIP-ViT-L/14 model in Fig. 6. As we analyzed previously in Section 4.3, we can confirm that the conditions for effective ensemble construction are met here as well, i.e., ensemble members exhibit slightly higher individual negative log-likelihoods (circle markers) compared to the pre-trained model (star-shaped marker) in the first subplot, and they also offer diverse function outputs, as shown in the subsequent subplots. As depicted in the leftmost

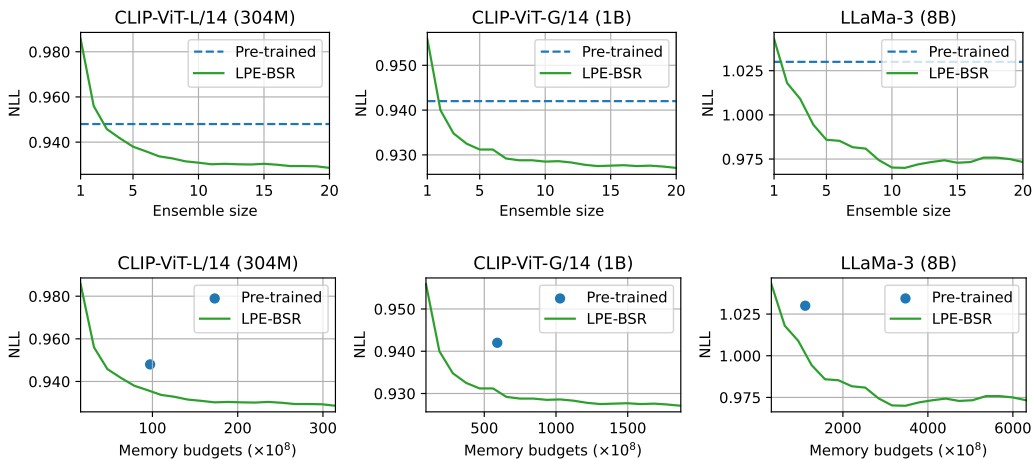

**Figure 7: Constructing low precision ensemble of large models.** Negative log-likelihood for pre-trained models (Pre-trained, shown in blue) and low precision ensembling with Bernoulli stochastic rounding centered around the pre-trained model (LPE-BSR, shown in green). The evaluation was conducted on ImageNet for CLIP models and on MMLU for LLaMa-3 in a zero-shot setting. **Top:** When the x-axis represents the ensemble size. **Bottom:** When the x-axis represents memory budgets, i.e., the total number of bits for representing ensemble.

**Table 3: Results for low precision ensembling of pre-trained models.** We compute (a) average loss, (b) ambiguity, and (c) ensemble loss for diversity analysis, along with evaluation metrics to assess overall performance. Our LPE-BSR samples are centered around ☆ within each group (pre-trained model in this context), which are separated by horizontal lines.

| Model | # Params | Method | System | Diversity analysis | | | Evaluation metrics | | |
|---|---|---|---|---|---|---|---|---|---|
| | | | | (a) | (b) | (c) | NLL | ERR | ECE |
| CLIP-ViT-L/14 | 0.30B | ☆ Pre-trained | FP32 | .948 | - | .948 | .948 | .251 | .049 |
| | | LPE-BSR | INT-5 | .993 | .053 | .940 | **.929** | **.250** | **.028** |
| CLIP-ViT-G/14 | 1.01B | ☆ Pre-trained | FP32 | .942 | - | .942 | .942 | **.206** | .095 |
| | | LPE-BSR | INT-5 | .955 | .013 | .941 | **.927** | **.206** | **.089** |
| LLaMa-3 | 8.03B | ☆ Pre-trained | FP32 | 1.03 | - | 1.03 | 1.03 | **.361** | .160 |
| | | LPE-BSR | INT-5 | 1.07 | .049 | 1.02 | **.923** | .364 | **.087** |

subplot of Fig. 7 it leads to achieving lower negative log-likelihood through LPE-BSR compared to the pre-trained model without the need for additional training.

Fig. 7 demonstrates that LPE-BSR consistently improves upon the pre-trained checkpoint, even for larger models such as CLIP-ViT-G/14 and LLaMa-3. The subplots at the top of Fig. 7 demonstrate that the performance of LPE-BSR improves with increasing ensemble size, while the subplots at the bottom show that LPE-BSR occupies the preferred lower-left region of the trade-off plots for memory budgets and performance, surpassing the pre-trained checkpoint. Table 3 offers more detailed results for an ensemble size of $S = 20$, including diversity analysis and evaluation results, further confirming the effectiveness of our proposed LPE-BSR for models with billions of parameters.

## 5 Related Work

The field of Bayesian deep learning provides the most relevant research for our low precision ensembling strategy; Ferianc et al. (2021) integrated quantization-aware training (Jacob et al., 2018) into established Bayesian deep learning methods; Zhang et al. (2022) introduced a technique for implementing stochastic gradient Langevin dynamics (Welling and Teh, 2011) with reduced precision. However, our work differs significantly from theirs in two key aspects: 1) They focused on

training from scratch, which deviates somewhat from the prevalent practice of utilizing pre-trained large models. 2) They employed small-scale models; the largest model they considered, ResNet-18 with 11 million parameters, falls outside our scope as discussed in Section 4.1, as we are interested in larger scales. Nonetheless, the interest in employing low precision ensembling in Bayesian deep learning holds significant promise. Our demonstration of its feasibility for large models constitutes a meaningful advancement for the Bayesian deep learning community (Papamarkou et al., 2024).

## 6   Conclusion

We provided a novel insight on ensembling within low precision number systems. While conventional wisdom perceives quantization errors stemming from representing neural network weights in low precision as obstacles, we introduced an innovative viewpoint suggesting that these errors could serve as a source of ensemble diversity. Our empirical results demonstrated that low precision weights obtained through stochastic rounding of pre-trained weights could effectively form ensembles and improve uncertainty estimates and calibration, especially for large models. Considering the growing scale of models in recent trends reduces the appeal of ensemble methods due to their inherent scalability issue, where memory costs increase with the number of ensemble components, our exploration of low precision ensembling lays the foundation for developing efficient ensemble methods in the era dominated by large models.

**Limitations and future directions.** At present, our investigations have centered on the simplest form of low precision number system, known as the symmetric uniform quantization scheme. Similar to the quest in neural network quantization for systems that yield better quantized solutions (e.g., Yvinec et al., 2023; Dettmers et al., 2024), the search for systems conducive to more effective low precision ensembling presents an intriguing avenue for future research. Furthermore, we used fake quantization across all experiments for research purposes, which prevented us from benchmarking the latency of the low precision ensemble due to limited access to specialized hardware and software for accelerating the inference speed of quantized models. Nonetheless, as our work relies on the standard symmetric uniform quantization scheme, it remains compatible with ongoing and future advancements in neural network quantization. Developing practical components such as custom CUDA kernels tailored to low precision ensembles would also be a promising future direction.

**Broader impacts.** Our method advocates for the utilization of large models, which could potentially raise ethical concerns (e.g., Weidinger et al., 2021). However, it is important to note that this work primarily focuses on analytical aspects and does not inherently entail significant ethical risks.

## Acknowledgement

This work was partly supported by Institute of Information & communications Technology Planning & Evaluation(IITP) grant funded by the Korea government(MSIT) (No.RS-2019-II190075, Artificial Intelligence Graduate School Program(KAIST), No.2022-0-00184, Development and Study of AI Technologies to Inexpensively Conform to Evolving Policy on Ethics, No.RS-2024-00509279, Global AI Frontier Lab), and National Research Foundation of Korea(NRF) grant funded by the Korea government(MSIT) (NRF-2021M3E5D9025030). This material is based upon work supported by the Google Cloud Research Credits program with the award GCP19980904 and Cloud TPUs from Google's TPU Research Cloud (TRC).

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

## A  Models and datasets

The pre-trained weights utilized in our experiments are listed below. We refer readers to the respective papers fdetails on each model: ViT (Dosovitskiy et al., 2021), CLIP (Radford et al., 2021; Cherti et al., 2023), and LLaMa (Touvron et al., 2023). In our CLIP experiments, we obtained zero-shot head weights by following the standard procedure outlined in the official code base[1].

- ViT-T/16: https://huggingface.co/WinKawaks/vit-tiny-patch16-224/tree/main
- ViT-S/16: https://huggingface.co/WinKawaks/vit-small-patch16-224
- ViT-B/16: https://huggingface.co/google/vit-base-patch16-224
- ViT-L/16: https://huggingface.co/google/vit-large-patch16-224
- CLIP-ViT-L/14: https://huggingface.co/openai/clip-vit-large-patch14
- CLIP-ViT-G/14: https://huggingface.co/laion/CLIP-ViT-bigG-14-laion2B-39B-b160k
- LLaMa-3-8B: https://huggingface.co/meta-llama/Meta-Llama-3-8B-Instruct

We employed two datasets for our experiments: ImageNet (Russakovsky et al., 2015) for ViT and CLIP-ViT models, and MMLU (Hendrycks et al., 2021) for LLaMa. The evaluation of MMLU was conducted using the template provided in the official repository[2], and the computation was based on a micro-average.

## B  Evaluation metrics

Let $p_i \in [0,1]^K$ represent the predicted categorical probabilities and $y_i \in \{1, \ldots, K\}$ denote the ground truth label for the $i^{\text{th}}$ data point, where the total number of data points is $N$.

**NLL.** The negative log-likelihood (NLL) of a categorical distribution, also known as cross-entropy loss, is a fundamental metric for evaluating the performance of a classification model:

$$\text{NLL} = \frac{1}{N} \sum_{i=1}^{N} \log p_i^{(y)}. \tag{6}$$

**ERR.** Another primary metric commonly used to assess the performance of a classification model is the classification error (ERR), also referred to as the 0-1 loss:

$$\text{ERR} = \frac{1}{N} \sum_{i=1}^{N} \left[ y \neq \arg\max_k p_i^{(k)} \right], \tag{7}$$

where $[\cdot]$ denotes the Iverson bracket.

**ECE.** The common choice for measuring calibration in machine learning is the expected calibration error (ECE), particularly its empirical variant with binning (Pakdaman Naeini et al., 2015):

$$\text{ECE} = \sum_{j=1}^{J} \frac{|B_j| \cdot |\text{acc}(B_j) - \text{conf}(B_j)|}{N}, \tag{8}$$

where $B_j$ denotes the $j^{\text{th}}$ bin comprising $|B_j|$ data points whose prediction confidence $\max_k p_i^{(k)}$ falls within the interval $((j-1)/J, j/J]$. Here, $\text{acc}(B_j)$ is the classification accuracy of $B_j$, and $\text{conf}(B_j)$ is the average confidence within $B_j$. As a result, it computes a weighted average of calibration gaps, which are the differences between accuracy and confidence, across bins. Throughout our experiments, we employed $J = 15$ bins for ECE computation.

**Ensemble ambiguity.** Let $z_{s,i} \in \mathbb{R}^K$ be categorical logits predicted by the $s^{\text{th}}$ model for the $i^{\text{th}}$ data point. The generalized ambiguity decomposition can be written as

$$\underbrace{\text{AMB}}_{\text{(b) ambiguity}} = \underbrace{\frac{1}{S} \sum_{s=1}^{S} \frac{1}{N} \sum_{i=1}^{N} \log \sigma(z_{s,i})^{(y)}}_{\text{(a) average loss}} - \underbrace{\frac{1}{N} \sum_{i=1}^{N} \log \sigma\left(\frac{1}{S} \sum_{s=1}^{S} z_{s,i}\right)^{(y)}}_{\text{(c) ensemble loss}}, \tag{9}$$

---

[1] https://github.com/openai/CLIP
[2] https://github.com/hendrycks/test

**Table 4: Motivating results for low precision ensembling of pre-trained ViT models.** Negative log-likelihood (NLL), classification error (ERR), and ensemble ambiguity (AMB) for rounding-to-nearest (RTN) and low precision ensembling with Bernoulli stochastic rounding (LPE-BSR) derived from the publicly available pre-trained ImageNet model (☆).

| Method | System | ViT-T/16 (6M) | | | ViT-S/16 (22M) | | |
|---|---|---|---|---|---|---|---|
| | | NLL | ERR | AMB | NLL | ERR | AMB |
| ☆ Pre-trained | FP32 | .932 | .243 | - | .667 | .185 | - |
| RTN | INT-6 | $.948_{\pm.001}$ | $.247_{\pm.001}$ | - | $.671_{\pm.001}$ | $.185_{\pm.001}$ | - |
| | INT-4 | $1.23_{\pm.001}$ | $.315_{\pm.001}$ | - | $.822_{\pm.001}$ | $.218_{\pm.001}$ | - |
| LPE-BSR | INT-6 | $.932_{\pm.001}$ | $.245_{\pm.001}$ | $.024_{\pm.001}$ | $.665_{\pm.001}$ | $.185_{\pm.001}$ | $.014_{\pm.001}$ |
| | INT-4 | $1.30_{\pm.001}$ | $.298_{\pm.001}$ | $.489_{\pm.001}$ | $.821_{\pm.001}$ | $.211_{\pm.001}$ | $.268_{\pm.001}$ |

| Method | System | ViT-B/16 (87M) | | | ViT-L/16 (307M) | | |
|---|---|---|---|---|---|---|---|
| | | NLL | ERR | AMB | NLL | ERR | AMB |
| ☆ Pre-trained | FP32 | .687 | .182 | - | .639 | .165 | - |
| RTN | INT-6 | $.687_{\pm.000}$ | $.182_{\pm.000}$ | - | $.639_{\pm.000}$ | $.165_{\pm.000}$ | - |
| | INT-4 | $.716_{\pm.001}$ | $.184_{\pm.001}$ | - | $.647_{\pm.001}$ | $.167_{\pm.000}$ | - |
| LPE-BSR | INT-6 | $.681_{\pm.001}$ | $.181_{\pm.001}$ | $.006_{\pm.000}$ | $.632_{\pm.001}$ | $.164_{\pm.001}$ | $.003_{\pm.000}$ |
| | INT-4 | $.648_{\pm.001}$ | $.175_{\pm.000}$ | $.088_{\pm.001}$ | $.600_{\pm.002}$ | $.160_{\pm.001}$ | $.037_{\pm.001}$ |

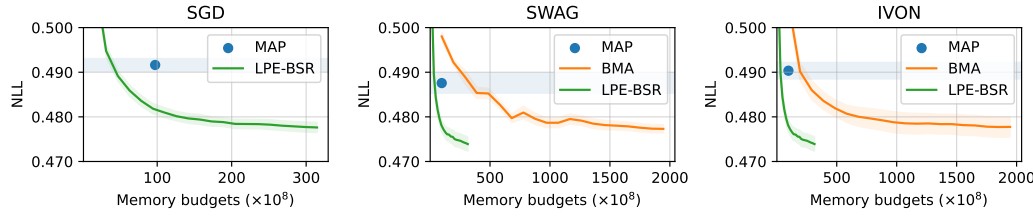

**Figure 8: Comparing low precision ensembling to Bayesian methods.** Negative log-likelihood for Bayesian model averaging using an approximate Gaussian posterior derived from SWAG or IVON (BMA, shown in orange) and low precision ensembling with Bernoulli stochastic rounding centered around the MAP solution obtained by each optimizer (LPE-BSR, shown in green).

where $\sigma$ denotes a softmax function that maps categorical logits into probabilities. It is worth noting that logit ensembling in (c) is essentially the same as computing a a normalized geometric mean for categorical probabilities. For further information, please see Wood et al. (2023).

# C  Additional results

## C.1  Motivating results with error bars

Table 4 is an extended version of Table 1, including standard deviations across four trials.

## C.2  Comparative results with Bayesian methods in terms of memory budgets

Fig. 8 is a modified version of Fig. 2, where the x-axis has been changed from ensemble size to memory budget, defined as the total number of bits used to represent the ensembles. It can be interpreted as trade-off plots between memory budgets and performance. Compared to SWAG and IVON, which represent ensemble members in the FP32 system, LPE-BSR occupies the preferred lower-left region, where ensemble members are represented in an INT-5 system.

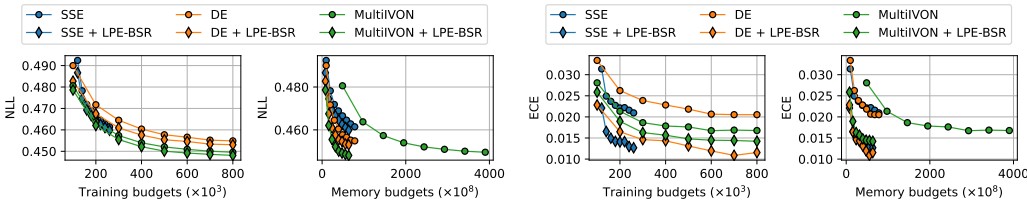

**Figure 9: Combining with ensembling methods.** Negative log-likelihood and expected calibration error for ensembling methods, SSE, DE, and MultiIVON, in terms of training budgets, i.e., the number of backward passes, and memory budgets, i.e., the total number of bits for representing ensemble. Here, DE represents an ensemble of multiple Adam solutions, while MultiIVON represents an ensemble of multiple IVON solutions.

**Table 5: Results for low precision ensembling of fine-tuned models.** We compute (a) average loss, (b) ambiguity, and (c) ensemble loss for diversity analysis, along with evaluation metrics to assess overall performance. The results are presented in ascending order of memory budgets, i.e., the total number of bits for representing ensemble. The number in parentheses after each method indicates the ensemble size.

| Method | System | Diversity analysis | | | Evaluation metrics | | | Memory budgets ($\times 10^8$) |
| | | (a) | (b) | (c) | NLL | ERR | ECE | |
| --- | --- | --- | --- | --- | --- | --- | --- | --- |
| LPE-BSR (4) | INT-5 | .513 | .025 | .488 | .481 | .138 | .021 | 62.9 |
| LPE-BSR (6) | INT-5 | .513 | .028 | .485 | .477 | .137 | .020 | 94.4 |
| ☆ MAP (1) | FP32 | .487 | - | .487 | .487 | .136 | .035 | 97.3 |
| BE (4) | FP32 | .492 | .006 | .486 | .480 | .137 | .032 | 97.4 |
| LPE-BSR (8) | INT-5 | .513 | .029 | .483 | .475 | .137 | .019 | 126. |
| DE (4) | FP32 | .488 | .020 | .468 | .462 | .132 | .026 | 389. |

## C.3 Further comparisons with non-Bayesian ensembles

While the proposed LPE-BSR method does not involve fine-tuning, though we do include fine-tuning experiments to compare LPE-BSR's ensemble quality with Bayesian methods in Section 4.2, a comparative study with other non-Bayesian ensemble techniques like *deep ensembles* (DE; Lakshminarayanan et al., 2017) and *batch ensembles* (BE; Wen et al., 2020) would be valuable.

Table 5 summarizes our experimental results using the Adam optimizer (Kingma and Ba, 2015). We observed that in LPE-BSR, (a) each ensemble member had relatively lower performance (= 0.513). However, (b) due to high ensemble diversity ($\geq 0.025$), (c) there was a significant improvement in the final ensemble performance. Consequently, it achieves performance comparable to BE, another memory-efficient method available in fine-tuning scenarios. In BE, ensemble members are similarly centered around one solution, with members derived from shared weights by multiplying rank-one matrices, while LPE-BSR members are derived from center weights using stochastic rounding. This comparison with BE, a well-known memory-efficient ensembling strategy, highlights the potential of low precision ensembling with LPE-BSR.

## C.4 Further comparisons with other training-free baselines

Our LPE-BSR method forms a low precision ensemble from a given checkpoint in a training-free manner, as presented in Section 4. We further conduct additional comparative experiments with two baselines: 1) *Gaussian*, which builds an ensemble by adding Gaussian noise with fixed variance to the pre-trained weights; and 2) *Monte Carlo Dropout* (MCD; Gal and Ghahramani, 2016), which constructs an ensemble by applying the dropout technique during inference. MCD is particularly relevant as it uses a $q(\boldsymbol{w})$ form similar to Eq. 5 of LPE-BSR, employing $\delta(\boldsymbol{0})$ and $\delta(\boldsymbol{w})$.

Table 6 summarizes the results for CLIP-ViT-L/14 with an ensemble size of $S = 20$. It clearly shows that while both the Gaussian and MCD baselines can perform ensembling in a training-free manner with appropriately tuned noise scales—specifically, the variance of Gaussian noise for the Gaussian

**Table 6: Comparative results for training-free ensembles.** The training-free ensemble methods, including Gaussian, MCD, and our proposed LPE-BSR, collect ensemble members centered around ☆ (pre-trained CLIP-ViT-L/14 model in this context). Here, $\sigma^2$ denotes the variance of Gaussian noise in the Gaussian baseline, and $p$ refers to the drop probability in the MCD baseline.

| Method | System | NLL | ERR | ECE |
|---|---|---|---|---|
| ☆ Pre-trained | FP32 | .948 | .251 | .049 |
| Gaussian ($\sigma^2 = 0.0001$) | FP32 | .948 | .251 | .049 |
| Gaussian ($\sigma^2 = 0.0002$) | FP32 | .948 | .251 | .048 |
| Gaussian ($\sigma^2 = 0.0004$) | FP32 | .946 | **.250** | .046 |
| Gaussian ($\sigma^2 = 0.0008$) | FP32 | .941 | **.250** | .043 |
| Gaussian ($\sigma^2 = 0.0016$) | FP32 | .934 | **.250** | .031 |
| Gaussian ($\sigma^2 = 0.0032$) | FP32 | .981 | .264 | .038 |
| MCD ($p = 0.001$) | FP32 | .946 | .251 | .047 |
| MCD ($p = 0.002$) | FP32 | .946 | .251 | .047 |
| MCD ($p = 0.004$) | FP32 | .944 | **.250** | .046 |
| MCD ($p = 0.008$) | FP32 | .940 | **.250** | .044 |
| MCD ($p = 0.016$) | FP32 | .938 | .251 | .041 |
| MCD ($p = 0.032$) | FP32 | .944 | .255 | .034 |
| LPE-BSR | INT-5 | **.929** | **.250** | **.028** |

baseline and the drop probability for the MCD baseline—our proposed LPE-BSR outperforms them. It is worth noting that LPE-BSR is more memory-efficient, as each of its ensemble members uses INT-5, compared to FP32 used by the baseline methods. Therefore, LPE-BSR not only achieves better performance but also does so with reduced memory usage.

# D   Experimental details

We built our experimental code using JAX (Bradbury et al., 2018) and Transformers (Wolf et al., 2020), both licensed under Apache-2.0.[3] We conducted experiments using TPUv2/v3/v4 cores, with flexibility in selecting the cores based on the memory requirements of each experiment. The code is available at https://github.com/cs-giung/lpe-bsr.

# E   Optimization and sampling methods

The optimization process for CLIP-ViT-L/14 models concludes after 100,000 iterations with a mini-batch size of 64, employing a cosine decaying learning rate schedule. In the experiments described in the main text, we employed the following optimizers: SGD, IVON, and SWAG. All hyperparameter tuning was conducted using a development set created by taking 1% of the training dataset, i.e., `'training[99%:]'` in TensorFlow Datasets (Abadi et al., 2015).

The zero-shot head weights were kept entirely fixed, meaning they were not fine-tuned or quantized in any of the experiments. We also froze the embedding layer to enable basic SGD update rules to function with the transformer architecture, as described by Kumar et al. (2024).

**SGD.** Stochastic gradient descent (SGD) is a foundational stochastic optimization algorithm in machine learning (Robbins and Monro, 1951). Although it is usually deemed ineffective for transformer architectures, recent findings by Kumar et al. (2024) has shown that a simple modification—freezing the embedding layer—enables pure SGD, even without momentum, to produce results competitive with Adam (Kingma and Ba, 2015). The hyperparameter settings for the SGD optimizer utilized in our experiments are summarized in Table 7.

**IVON.** Efforts to develop variational methods for implementing Bayesian inference on neural network models have continued over time (Graves, 2011; Blundell et al., 2015). However, these at-

---

[3]https://www.apache.org/licenses/LICENSE-2.0

**Table 7: Hyperparameters in SGD and IVON.**

| Method | Hyperparameter | Search space |
|---|---|---|
| SGD | Base learning rate $\alpha_0 = 0.003$ | $\{0.03, 0.01, 0.003, 0.001\}$ |
| | $\ell_2$ regularization $\delta = 10^{-3}$ | $\{10^{-2}, 10^{-3}, 10^{-4}, 10^{-5}\}$ |
| IVON | Base learning rate $\alpha_0 = 0.003$ | $\{0.03, 0.01, 0.003, 0.001\}$ |
| | Effective sample size $\lambda = 1268355$ | - |
| | $\ell_2$ regularization $\delta = 10^{-4}$ | $\{10^{-2}, 10^{-3}, 10^{-4}, 10^{-5}\}$ |
| | Gradient momentum $\beta_1 = 0.9$ | - |
| | Hessian momentum $\beta_2 = 0.99999$ | $\{0.9999, 0.99999, 0.999999\}$ |
| | Hessian initialization $h_0 = 1$ | - |
| | Clip radius $\xi = 10^{-3}$ | $\{10^{-2}, 10^{-3}, 10^{-4}\}$ |

tempts have frequently proven ineffective in practical scenarios, even for moderately-sized problems (Osawa et al., 2019). Recently, the Improved Variational Online Newton (IVON) algorithm, introduced by Shen et al. (2024), has facilitated variational learning for large models with an update rule closely resembling that of Adam (Kingma and Ba, 2015). In essence, by modifying the update rule of the second momentum in Adam, IVON estimates a diagonal covariance of an approximate Gaussian posterior. The hyperparameters employed for the IVON optimizer in our experiments are presented in Table 7; the notations adhere to those described in Shen et al. (2024).

**SWAG.** Besides variational learning, another notable approach for obtaining an approximate Gaussian posterior is Stochastic Weight Averaging Gaussian (SWAG; Maddox et al., 2019). Essentially, SWAG entails collecting samples throughout the SGD update process and calculating their sample mean and covariance to approximate the Gaussian posterior. For SWAG, we initially applied SGD updates with a cosine decay schedule until reaching a non-zero constant learning rate over 80,000 iterations. Subsequently, we switched to constant learning rate SGD updates, collecting samples every 1,000 iterations. The constant learning rate is determined by multiplying the base learning rate by a decaying factor $\lambda_{\text{SWAG}} \in \{1.0, 0.5, 0.2, 0.1\}$, and the final hyperparameter configuration was $\alpha_0 = 0.003$ and $\lambda_{\text{SWAG}} = 0.2$. Ultimately, SWAG approximates a Gaussian posterior with non-diagonal covariance matrix with a rank of 10 from these 20 samples.

**SSE and CSGLD.** When employing the SGD optimizer, the only difference between Snapshot Ensembling (SSE; Huang et al., 2017) and Cyclical Stochastic Gradient Langevin Dynamics (CSGLD; Zhang et al., 2020) lies in the incorporation of Gaussian noise in the update rule; in CSGLD, a noise term derived from stochastic gradient Langevin dynamics (Welling and Teh, 2011) is added at each iteration. In our experiments on fast ensembling, we initialized both SSE and SGLD using the outcomes obtained from training with SGD for 100,000 iterations. Employing a cosine-decaying learning rate schedule with a cycle duration of 20,000 iterations, we iterated this schedule ten times to produce a total of ten snapshots.

