# OpenReview forum: "Ex Uno Pluria: Insights on Ensembling in Low Precision Number Systems"
_NeurIPS.cc/2024/Conference — NeurIPS 2024 poster_

### Official Review · Reviewer_svQu · 2024-06-13

**Soundness:** 3
**Presentation:** 3
**Contribution:** 2
**Rating:** 4
**Confidence:** 3

**Summary:**

This paper addresses the problem of ensembling in low precision number systems, where quantized models act as members of the ensemble. The authors suggest that quantization errors can be leveraged to enhance ensemble diversity. Based on this concept, they propose a method called LPE-BSR. Through extensive experiments and analyses, the authors validate the effectiveness of their approach.

**Strengths:**

1. The paper is well-organized and clearly written.
2. The authors perform extensive experiments and analyses regarding ensembling in low precision number systems.

**Weaknesses:**

1. The paper relies on existing techniques for both ensembling and quantization. It primarily presents an empirical study on the combination of these two approaches rather than introducing a new method.
2. The practical utility of the proposed LPE-BSR method is somewhat constrained. Large models typically incur high inference costs, and ensembling them—even with low precision systems—can amplify these costs, particularly with ensemble sizes of 10 or 20, as discussed in the paper. Furthermore, according to the experimental results in Table 3, the performance improvement of the quantized ensemble over the full precision model is marginal. As an empirical study, the authors should better justify why ensembling in low precision number systems is a meaningful objective.
3. The related work section does not provide enough context or comparison with existing research. It would be beneficial to include a more detailed literature review.
4. The subfigures in Figures 3, 4, and 6 are difficult to interpret. Adding captions or labels for each subfigure would improve clarity.

**Questions:**

1. In line 174, what does $w_\text{MAP}$ mean? Is it a typo for $w^*_\text{MAP}$?
2. In line 179, the paper concludes that "it encountered difficulties in accurately estimating the diagonal covariance matrix." However, this conclusion is not clearly supported by the results presented in Figure 2. Please provide more evidence or clarification for this statement.

**Limitations:**

Yes.

---

> ### Author Rebuttal · Authors · 2024-08-06
>
> Thank you for your insightful review. We hope our response below addresses any remaining concerns. If you have any further questions, please let us know. Otherwise, we kindly request that you reconsider your assessment accordingly. Thank you again for your valuable feedback!
>
> ---
>
> > In line 174, what does $w_{\text{MAP}}$ mean? Is it a typo for $w_{\text{MAP}}^\ast$?
>
> Apologies for the confusion caused by the typos. We have identified that asterisks were missing in lines 174 and 232. Thank you for your meticulous review!
>
> > In line 179, the paper concludes that "it encountered difficulties in accurately estimating the diagonal covariance matrix." However, this conclusion is not clearly supported by the results presented in Figure 2. Please provide more evidence or clarification for this statement.
>
> We initially concluded this because the SWAG-Diag baseline seemed ineffective at performing BMA. Without knowing the true posterior, we assessed the quality of the approximate Gaussian posterior through MAP and BMA performance. Good MAP performance suggested the mean was acceptable, but poor BMA indicated a problem with the variance estimate. However, after conducting additional experiments, we improved the SWAG baseline results, as detailed in the General Response. This resolves the issue with the SWAG baseline, which will be updated in the camera-ready version. Thank you for raising concerns about the SWAG results.
>
> > The paper relies on existing techniques for both ensembling and quantization. It primarily presents an empirical study on the combination of these two approaches rather than introducing a new method.
>
> Our main contribution is offering a fresh perspective by using quantization errors--often seen as challenges in quantization research--as a source of diversity in ensemble methods. Consequently, although each individual approach (quantization and ensemble) may not be novel by itself, we believe showcasing the potential of low-precision ensembles for modern large-scale models represents a novel and significant contribution.
>
> > The practical utility of the proposed LPE-BSR method is somewhat constrained. Large models typically incur high inference costs, and ensembling them—even with low precision systems—can amplify these costs, particularly with ensemble sizes of 10 or 20, as discussed in the paper. Furthermore, according to the experimental results in Table 3, the performance improvement of the quantized ensemble over the full precision model is marginal. As an empirical study, the authors should better justify why ensembling in low precision number systems is a meaningful objective.
>
> As you noted, the practicality of ensembling large models is constrained, which we believe explains the decreased preference for ensemble methods in the current era of large models. Our research addresses this by demonstrating that scalability issues with ensemble techniques can potentially be mitigated through the use of low precision number systems, particularly for modern large models. In doing so, we aim to rekindle interest in ensemble methods within the context of current large model trends, as mentioned in lines 289-293 of the conclusion section. We hope that our work on LPE-BSR will be seen as a pioneering effort in showcasing the potential of low precision ensembles.
>
> Besides, following Reviewer 9d5B's recommendation, we have extended the memory budgets plot from Section 4.4 to include the experiments from Sections 4.2 and 4.5. The figures in the attached PDF demonstrate that our LPE-BSR approach is located in the lower-left corner, signifying that it achieves lower NLL while being more memory-efficient. Even with a smaller ensemble size of five, it outperforms both the BMA baselines and the original checkpoint.
>
> > The related work section does not provide enough context or comparison with existing research. It would be beneficial to include a more detailed literature review.
>
> Thank you for your constructive comment, which strengthens our paper! We have reviewed relevant studies on low precision ensembling, especially those involving Bayesina methods with quantization-aware training, but have struggled to find further research on this topic. Any recommendations for further literature would be immensely helpful. Meanwhile, we will revise the paper to address your feedback by expanding the related work section to include additional paragraphs on ensemble methods and quantization.
>
> > The subfigures in Figures 3, 4, and 6 are difficult to interpret. Adding captions or labels for each subfigure would improve clarity.
>
> Even though lines 210-216 of the main text explain the radial landscape plots, it appears that Reviewer bKfp also expressed similar concerns about Figure 3. Including captions and labels for each figure, as you recommended, should improve readability in the camera-ready version. We appreciate your constructive feedback!

---

> > ### Comment · Reviewer_svQu · 2024-08-13
> >
> > Thank you for the response. Most of my questions have been addressed. However, regarding the inference costs, memory budget is not the only factor need to be take into consideration. The inference latency should also be compared, since the ensemble of plenty of models would greatly increase the computational complexity.

---

> > > ### Author Response · Authors · 2024-08-13
> > >
> > > We acknowledge the reviewer’s concerns about the inherent memory and speed costs of ensemble methods. However, we want to highlight that our low-precision ensembling approach can significantly alleviate these issues through the use of quantized models.
> > >
> > > One of the key advantages of our proposed low-precision ensembling method is that the reduced size of each quantized model makes parallel execution more feasible. Unlike conventional ensembles of full-precision models, which often struggle with parallelization due to their large memory footprint and must be executed sequentially, our approach allows for efficient parallelization. For example, our LPE-BSR (INT-5) ensemble, consisting of five models, requires less memory than a single FP32 model. This means that in environments where full-precision FP32 models are deployable, there is sufficient memory to support the parallel execution of our LPE ensemble.
> > >
> > > While parallelization can significantly boost efficiency, we acknowledge that it may not entirely eliminate the latency issues associated with using multiple models in an ensemble. To better address these latency challenges, it is essential to utilize specialized hardware and software optimized for accelerating the inference of quantized models. Although we used fake quantization for our research due to the lack of access to such hardware, our method follows the standard symmetric uniform quantization scheme, which is compatible with both existing and emerging advancements in neural network quantization. By integrating cutting-edge techniques from the quantization community, we can further improve the inference speed of parallelized ensembles and more effectively address latency concerns.
> > >
> > > We assure you that all these points will be thoroughly addressed in the final manuscript. We welcome any additional comments and suggestions, and we appreciate your constructive feedback and the opportunity to enhance our work.
> > >
> > > Sincerely,
> > > The authors of Submission 19991

---

> ### Comment · Reviewer_svQu · 2024-08-13
>
> I appreciate the authors' prompt response. As mentioned, the LPE-BSR (INT-5) ensemble model does outperform a single FP32 model in terms of both memory footprint and performance. However, if we were to simply adopt a model with more parameters and better performance, applying standard quantization, it might lead to a single dense model with similar memory requirements while performing better than the ensemble. Additionally, dense models typically benefit from a higher level of parallelism in practice compared to an ensemble approach. Therefore, the practical effectiveness of the proposed method is not sufficiently convincing for me. Given this, I will maintain my initial score.

---

> > ### Author Response · Authors · 2024-08-14
> >
> > Our approach indicates that forming a low-precision ensemble from a pre-trained model can improve performance over the original. While there is some debate about whether a quantized version of a larger pre-trained model should be considered--since such models are not always available--we want to emphasize that our work, which highlights the potential of low-precision ensembling for large modern models, remains significant and relevant. Thank you again for taking the time to review our paper!

---

### Official Review · Reviewer_bKfp · 2024-07-10

**Soundness:** 3
**Presentation:** 3
**Contribution:** 3
**Rating:** 5
**Confidence:** 4

**Summary:**

This paper presents a new way to generate an ensemble of models without the need for training multiple times and with the extra advantage of using low-precision representation which inherently saves memory.
The idea is to build an ensemble of models starting from stochastic variations of a single model. Those variations are based on a stochastic rounding scheme that samples either the ciel or the floor of a real number proportionally to the distance to the two rounded values (called Bernoulli stochastic rounding).
Results show that, especially for large models (see Tab.1), this technique provides accuracies that are better than a single model with the advantage of a quantized representation that has a lower memory footprint.

**Strengths:**

- The proposed idea is interesting: how to use the noise generated by quantisation as a way to generate diversity for ensembling.

- The presentation of the paper is normally good although the experiments are a bit misleading (see below)

**Weaknesses:**

\- The proposed idea is interesting and I think it has some potential. However, I think that the authors did not analyse and compare in a fair way the drawbacks of the approach. Below some points.

\- Compare with Deep ensembling. The method is interesting because it does not require to retrain the model. However, authors should compare with a real ensembling of multiple models.

\- Compare with others without adding its contribution. If the pre-training of large models is too expensive to be evaluated multiple times, the authors should compare with Bayesian and fast ensembling methods. Thus, instead of just adding the approach on top of SWAG, IVON and SSE, it would be interesting to see comparative results of each technique independently. For instance in Tab.2 we can see that in terms of error, the proposed approach (LPE-BSR) trained with normal SGD (error=.137) is inferior to IVON (error=.135) and comparable to SWAG (error=.137).

\- Weight averaging. Another important point when working with pre-trained models is weight averaging. As the different ensembles come from the same pre-training, normally they are aligned and the ensemble can be approximated with weight averaging. This allows to gain in accuracy with a single inference. What is the performance of the proposed approach with weight averaging?

\- Another possible baseline to compare with is to build an ensemble by adding Gaussian noise with a fixed variance for all values tuned (for ensembling) on a validation set. Although very simple, that approach can be quite effective.

**Questions:**

See the questions above associated to weaknesses.

Additional questions:

\- In tab.2 Why not showing all configs for SGD?

\- Fig.3, what is the difference among the three subfigures?

**Limitations:**

Some of the possible limitations of the approach are not analysed in the paper.
For instance, the proposed approach is not contrasted with other common ensembling techniques. Instead it is added to them. In this way, it is difficult to know the real efficacy of the approach.
Also, I did not see the performance of the approach for weight averaging. If the approach does not work well for weight averaging, it should be mentioned.

---

> ### Author Rebuttal · Authors · 2024-08-06
>
> Thank you for recognizing the potential and interest in our idea. We appreciate your detailed and constructive feedback. We have addressed your comments below and are confident that incorporating these revisions into the final version will significantly strengthen our work. If you have any additional concerns, please let us know. Otherwise, we kindly ask that you revise your assessment accordingly. Thank you again for your valuable feedback!
>
> ---
>
> > Another possible baseline to compare with is to build an ensemble by adding Gaussian noise with a fixed variance for all values tuned (for ensembling) on a validation set. Although very simple, that approach can be quite effective.
>
> We recognized that the mentioned baseline offers a simple approach for training-free ensemble construction. As a result, we included additional comparative experiments with the Gaussian baseline in the General Response, along with the MCDropout baseline mentioned by another reviewer. The results validate that LPE-BSR outperforms these baselines. Thank you for your constructive suggestion!
>
> > Compare with others without adding its contribution. (...)
>
> Our LPE-BSR method forms a low precision ensemble from a given checkpoint in a training-free manner, making it dependent on existing checkpoint. In our paper, we conducted a comparative study using fine-tuned checkpoints from SWAG, IVON, and SSE. Therefore, we respectfully disagree with the claim that SGD plus LPE-BSR should be compared to the MAP of IVON and SWAG; both IVON and SWAG are simply optimizers with additional features that provide an approximate Gaussian posterior. Specifically, SWAG is an enhanced version of SGD with Polyak-Ruppert averaging (cf. Izmailov et al., 2018), and IVON is a variation of Adam interpreted through variational learning (cf. Shen et al., 2024). As a result, it is entirely possible for the most basic optimizer, SGD without momentum (NLL = .492), to perform worse than the more advanced optimizers, SWAG and IVON (NLL = .488 and .489), and there is no reason why LPE-BSR should be limited to use with SGD. Importantly, LPE-BSR consistently improves performance over the given checkpoint (MAP), with improvements even surpassing those seen with BMA in IVON and SWAG ($\Delta$):.
>
> | Optimizer | Method  | NLL / ERR / ECE    | $\Delta$           |
> | :-        | :-      | :-                 | :-                 |
> | SGD       | MAP     | .492 / .138 / .035 | -                  |
> |           | LPE-BSR | .477 / .137 / .020 | .015 / .001 / .015 |
> | Adam      | MAP     | .487 / .136 / .035 | -                  |
> |           | LPE-BSR | __.469__ / __.135__ / __.019__ | .018 / .001 / .016 |
> | SWAG      | MAP     | .488 / .137 / .034 | -                  |
> |           | BMA     | .479 / .136 / .027 | .009 / .001 / .007 |
> |           | LPE-BSR | .473 / .136 / .021 | .015 / .001 / .013 |
> | IVON      | MAP     | .489 / .136 / .037 | -                  |
> |           | BMA     | .475 / __.135__ / .026 | .014 / .001 / .011 |
> |           | LPE-BSR | .472 / __.135__ / .023 | .017 / .001 / .014 |
>
> We have further included results using Adam as well as SGD in the table for readers who may not be familiar with the SWAG and IVON optimizers. Given that LPE-BSR generates low precision ensemble members around the provided checkpoint (MAP), it is anticipated that LPE-BSR will perform better with a more advanced optimizer, such as Adam with an NLL of 0.487. Again, the main point is that LPE-BSR reliably produces a low precision ensemble that exceeds the performance of the original FP32 solution without additional training, achieving quality on par with leading Bayesian methods like SWAG and IVON.
>
> > Compare with Deep ensembling. (...)
>
> Following the suggestions from Reviewers yx79 and bKfp, we compared LPE-BSR with deep ensembles (DE) and batch ensembles (BE). While LPE-BSR does not involve fine-tuning, these comparisons are valuable, similar to those with the BMA approach. Due to character limits, please see our response to Reviewer yx79. These comparisons show that LPE-BSR is a memory-efficient method achieving performance levels bounded by DE, similar to BE, but without requiring fine-tuning. Thank you for the constructive feedback!
>
> > Weight averaging. (...)
>
> In short, averaging the weights of LPE-BSR ensemble members results in the original central FP32 weights, similar to how averaging samples in SWAG and IVON results in the original Gaussian mean. Due to space constraints, please refer to our response regarding Model Soups to Reviewer 9d5B for more details.
>
> > Figure 3, what is the difference among the three subfigures?
>
> Details about radial basis plots are provided in lines 210-216. In short, the first subplot illustrates where each model is positioned within the loss landscape, while the next two subplots show how much the models are diverse from one another. Reviewer svQu also raised similar concerns regarding the clarity of the radial basis plots. Consequently, we intend to enhance the captions and labels in the camera-ready version. We appreciate your feedback on this matter.
>
> > Table 2, why not showing all configs for SGD?
>
> Thank you for bringing that to our attention. We initially removed the content to save vertical space and did not reinclude it. The missing results for SGD presented below, which show trends consistent with other results, will be added to the camera-ready version. We appreciate your thorough feedback!
>
> | System | (a)  | (b)  | (c)  | NLL  | ERR  | ECE  |
> | :-     | :-   | :-   | :-   | :-   | :-   | :-   |
> | INT-6  | .495 | .063 | .488 | .485 | .138 | .030 |
> | INT-5  | .513 | .057 | .456 | __.477__ | __.137__ | __.020__ |
> | INT-4  | .663 | .120 | .544 | .526 | .150 | .029 |

---

### Official Review · Reviewer_yx79 · 2024-07-11

**Soundness:** 3
**Presentation:** 3
**Contribution:** 2
**Rating:** 4
**Confidence:** 4

**Summary:**

This paper addresses the scalability challenge in ensembling deep neural networks for large models by introducing a novel low precision ensembling method. The approach generates an ensemble of models from a single model using low precision number systems in a training-free manner. Empirical analysis shows that this method achieves comparable or superior generalization performance. The findings suggest that low precision ensembling is a promising solution to enhance generalization performance while addressing the scalability issues inherent in traditional ensembling methods.

**Strengths:**

- The paper tackles an important problem on how to efficiently obtain an ensemble of models.
- The paper is well written and easy to follow.
- The proposed method is technically sound.
- The experimental results suggest that the proposed method is effective.

**Weaknesses:**

- Firstly, I think the paper is lacking some novelty. While interesting, it is not very surprising to me that an ensemble of randomly sampled low precision models can improve the generalization performance and uncertainty calibration. There have been numerous works of similar flavor.
- I think the paper lacks some important benchmark comparisons. For instance, how does the proposed method compare to other sampling based ensembling approaches like dropout ensemble [1] and batchnorm ensemble [2]?

[1] "Dropout as a Bayesian Approximation: Representing Model Uncertainty in Deep Learning"

[2] "Bayesian Uncertainty Estimation for Batch Normalized Deep Networks"

**Questions:**

- How does the diversity of low precision ensembles compare with other popular ensembling techniques like deep ensemble and batch ensemble?
- It would also be interesting to benchmark the performance of the proposed method in terms of computation cost (in addition to the memory budgets shown in Figure 5).

[1] "BatchEnsemble: An Alternative Approach to Efficient Ensemble and Lifelong Learning"

[2] "Simple and Scalable Predictive Uncertainty Estimation using Deep Ensembles"

**Limitations:**

Yes, the paper addresses the limitations of the work.

---

> ### Author Rebuttal · Authors · 2024-08-06
>
> Thank you for recognizing the significance of the problem we addressed, the clarity of our writing, the technical robustness of our proposed method, and the effectiveness of our experimental results. We have addressed your comments below and believe that incorporating these revisions into the final version will significantly enhance the contribution of our work. If you have any further concerns, please let us know. Otherwise, we kindly ask that you adjust your assessment accordingly. We appreciate your valuable feedback once again!
>
> ---
>
> > Firstly, I think the paper is lacking some novelty. While interesting, it is not very surprising to me that an ensemble of randomly sampled low precision models can improve the generalization performance and uncertainty calibration. There have been numerous works of similar flavor.
>
> We respectfully disagree with the novelty claim and would like to quote Michael Black’s remark: "If it is easy to explain and obvious in hindsight, this in no way diminishes the creativity (and novelty) of the idea." To our knowledge, the concept of employing quantization erros to obtain ensemble diversity has not been widely explored. Could you please provide specific examples of the "numerous works of similar flavor" you mentioned?
>
> > I think the paper lacks some important benchmark comparisons. For instance, how does the proposed method compare to other sampling based ensembling approaches like dropout ensemble [1] and batchnorm ensemble [2]?
>
> Thank you for your insightful comment! Both MCDropout (Gal and Ghahramani, 2016) and MCBatchNorm (Teye et al., 2018) leverage dropout and batch normalization techniques--typically used during training--at the inference stage to build ensembles. These methods indeed enable training-free ensemble construction, making them valuable for comparison with our proposed method. However, since modern transformer architectures typically do not include batch normalization layers, we concentrated our additional experiments on MCDropout. MCDropout is particularly relevant as it uses a $q(w)$ form similar to Eq. (5) of LPE-BSR, employing $\delta(0)$ and $\delta(w)$. The General Response provides additional comparative results with the MCDropout baseline and the Gaussian baseline mentioned by another reviewer. The results confirm that LPE-BSR outperforms both of these baselines. Thanks again for your constructive suggestion!
>
> > How does the diversity of low precision ensembles compare with other popular ensembling techniques like deep ensemble and batch ensemble?
>
> While our LPE-BSR method does not involve fine-tuning (though we do include fine-tuning experiments to compare LPE-BSR’s ensemble quality with Bayesian methods in our paper), a comparative study with other ensemble techniques like deep ensembles (DE) and batch ensembles (BE), as you suggested, would be valuable. Below are the results using the Adam optimizer; the number of ensemble members for each method is indicated in parentheses; the metrics (a), (b), and (c) correspond to those discussed in the diversity analysis of our paper; (a) average loss, (b) ambiguity (i.e., ensemble diversity), and (c) ensemble loss.
>
> | Method      | (a) / (b) / (c)    | NLL / ERR / ECE    | Memory Budget (x 1e8)
> | :- | :- | :- | :- |
> | DE (4)      | .488 / .020 / .468 | __.462__ / __.132__ / .026 | 389.
> | BE (4)      | .492 / .006 / .486 | .480 / __.137__ / .032 | 97.4
> | LPE-BSR (4) | .513 / .025 / .488 | .481 / .138 / .021 | __62.9__
> | LPE-BSR (6) | .513 / .028 / .485 | .477 / __.137__ / __.020__ | __94.4__
> | LPE-BSR (8) | .513 / .029 / .483 | __.475__ / __.137__ / __.019__ | 126.
>
> In our experiments using the Adam optimizer, we found that in LPE-BSR, (a) each ensemble member had relatively lower performance (NLL = 0.513). However, (b) due to high ensemble diversity (ambiguity ≥ 0.025), (c) there was a significant improvement in the final ensemble performance. Consequently, it achieves performance comparable to another memory-efficient method available in fine-tuning scenarios, BE. In BE, ensemble members are similarly centered around one solution, with BE members derived from shared weights by multiplying rank-one matrices, and LPE-BSR members derived from the center weights using stochastic rounding. This comparison with BE, a well-known memory-efficient ensembling strategy, highlights the potential of low precision ensembling with LPE-BSR.
>
> Furthermore, it is worth noting that LPE-BSR forms low precision ensembles without requiring additional training on any given checkpoint. This suggests that, similar to the enhancements shown in fast ensembling experiments, deep ensembles can also achieve further improvements with LPE-BSR. Figure 12 in the attached PDF illustrates this point clearly.
>
> > It would also be interesting to benchmark the performance of the proposed method in terms of computation cost (in addition to the memory budgets shown in Figure 5).
>
> We understand that you are addressing computation cost in terms of time complexity, such as latency, rather than space complexity like memory budgets. Unfortunately, quantized models typically have slower inference speeds without specialized hardware and software. As we do not have access to such hardware, we employed fake quantization for research purposes, which makes direct benchmarking challenging. However, since our work is based on the standard symmetric uniform quantization scheme, it aligns with current and future advancements in neural network quantization. We also want to highlight that our contribution, which showcases the potential of low-precision ensembling in large modern models, remains significant and relevant. We will include this point in the limitations section.

---

> > ### Comment · Reviewer_yx79 · 2024-08-13
> >
> > I would like to thank the authors for providing a careful feedback on my comments and additional experiments to demonstrate the effectiveness of the proposed method. Most of my concerns regarding comparisons with other benchmarks have been resolved. However, I still share the same reservation as reviewer svQu in terms of the novelty of the paper (that the proposed method is "an empirical study on the combination of these two approaches") and the "inference latency" (that the proposed method only addresses the memory cost issue but not inference latency). While I understand that it can be hard to do a thorough empirical investigation without specialized hardwares, qualitative discussions on the potential benefits the proposed quantization can bring is arguably much less convincing. As such, I would like to retain my original score.

---

> > > ### Author Response · Authors · 2024-08-14
> > >
> > > For the latency issue, as noted in our response to Reviewer svQu, it can be effectively managed through parallelization and techniques from the quantization field. For example, Kim et al. (2024) demonstrate improved latency with quantized models, with the RTN method we utilized showing the greatest speedup. Regarding the novelty aspect, we would appreciate it if you could provide the relevant works of a similar flavor you mentioned, as they would greatly assist in further refining our research. Thank you again for reviewing our paper!
> > >
> > > ---
> > > Kim et al. (2024), SqueezeLLM: Dense-and-Sparse Quantization.

---

### Official Review · Reviewer_9d5B · 2024-07-12

**Soundness:** 3
**Presentation:** 3
**Contribution:** 3
**Rating:** 6
**Confidence:** 4

**Summary:**

The paper proposes that ensembles of quantized low-precision instances of large models outperform the source models on image classification and MMLU tasks. The low precision models are generated using Bernoulli stochastic rounding. The authors support their claims by presenting empirical results for several models, and show improvements over bayesian ensembling methods on negative log likelihood, ensemble diversity and error rate.

**Strengths:**

1. The idea of using ensembles of low precision derivatives of existing pretrained large models is inspired. Recent works on linear mode ensembling and model soups have shown that trained parameters of a neural network, when perturbed often exhibit good ensembling properties. Using a quantization technique to reduce compute requirements of an ensemble while improving performance is an interesting approach.

2. The empirical results show improved performance for large model (ViT-B/16, Llama3) ensembles, and show good ensemble diversity.

3. The paper is well-written and easy to follow.

**Weaknesses:**

1. The paper presents NLL comparisons with SWAG-Diag and IVON. It would be great to see accuracy numbers as well to understand the actual performance differences as all the results are presented on classification tasks. In addition, Table. 1 should have error estimates for LPE-BSR given that it relies on randomness. Also given that LPE-BSR is not really a Bayesian method, it would be great to also compare with other ensembling approaches like Model Soups to see where it lands on the pareto curve of compute-vs-accuracy. Also, the SWAG algorithm shows significant performance differences between SWAG, and SWAG-diag. While it may not be possible to compare for all the presented models, an exemplar comparison would also help support the conclusions presented in the paper.

2. It would also be great to see some additional analysis on the actual compute savings incurred using LPE-BSR.

**Questions:**

1. Fig. 2 shows that SWAG does not improve with increasing ensembl sizes. This is a bit counter-intuitive. I may have misunderstood the figure, and invite some clarification if so.

2. Are the LLAMA results presented on MMLU 5-shot or 0-shot?

**Limitations:**

The authors discuss some limitations in the paper. However, I suggest adding the actual compute requirements for the ensemble of low precision models.

---

> ### Author Rebuttal · Authors · 2024-08-06
>
> We are pleased with the positive feedback that highlights our work as both inspired and interesting. We hope the responses provided below address any remaining concerns. Please let us know if there are any further issues. Thank you for your valuable comments!
>
> ---
>
> > The paper presents NLL comparisons with SWAG-Diag and IVON. It would be great to see accuracy numbers as well to understand the actual performance differences as all the results are presented on classification tasks.
>
> The ERR values presented in our paper represent classification error, computed as one minus the classification accuracy (i.e., ERR = 1.0 - ACC). We opted to report classification error instead of classification accuracy to ensure that our evaluation metrics (i.e., NLL, ERR, ECE) align with the “lower is better.” However, if using ACC enhances readability, we are willing to make that change in the final camera-ready version.
>
> > In addition, Table. 1 should have error estimates for LPE-BSR given that it relies on randomness.
>
> We empirically checked that the variability in LPE-BSR performance due to randomness is minimal (≤ 0.002). This is likely due to our experimental setup utilizing the same pre-trained model. We also agree that error bars should be included whenever possible, regardless of how small they are, and the camera-ready version will include results with error bars in the appendix. Thanks for the constructive feedback!
>
> > Also given that LPE-BSR is not really a Bayesian method, it would be great to also compare with other ensembling approaches like Model Soups to see where it lands on the pareto curve of compute-vs-accuracy.
>
> Model Soups (Wortsman et al., 2022) involves averaging the weights of models fine-tuned with different hyperparameters from the same pre-trained model. Thus, it is distinct from our LPE-BSR method, which does not involve any fine-tuning. Model Soups boosts the performance of the final fine-tuned checkpoint by averaging multiple fine-tuned checkpoints, whereas LPE-BSR works directly on a single provided checkpoint. Consequently, our LPE-BSR method can also be applied to a checkpoint obtained through Soup. Given that LPE-BSR generates low precision ensemble members around the provided checkpoint (such as MAP or Soup here), it is anticipated that LPE-BSR will achieve better performance with a more advanced checkpoint, like one from Soup with an NLL of .477. Notably, LPE-BSR consistently improves performance whether the given checkpoint comes from MAP or Soup, showcasing the flexibility of the proposed low precision ensembling method.
>
> | Method | NLL / ERR / ECE | $\Delta$ |
> | :- | :- | :- |
> | MAP (Adam) | .487 / .136 / .035 | - |
> | w/ LPE-BSR | __.469__ / __.135__ / __.019__ | __.018__ / __.001__ / __.016__ |
> | Soup (Adam) | .477 / .133 / .023 | - |
> | w/ LPE-BSR | __.465__ / __.132__ / __.017__ | __.012__ / __.001__ / __.006__ |
>
> In addition, while weight averaging is straightforward with standard FP32 weights, it becomes more complex with low precision number systems because the averaged points do not align with the low precision system grid. More importantly, regardless of the low precision grid, averaging the weights of LPE-BSR ensemble members results in the original central FP32 weights, similar to how averaging samples in SWAG and IVON results in the original Gaussian mean. We will clarify this point further in the final version of our paper.
>
> > Also, the SWAG algorithm shows significant performance differences between SWAG, and SWAG-diag. While it may not be possible to compare for all the presented models, an exemplar comparison would also help support the conclusions presented in the paper.
>
> > Fig. 2 shows that SWAG does not improve with increasing ensembl sizes. This is a bit counter-intuitive. I may have misunderstood the figure, and invite some clarification if so.
>
> We are happy to share the improved SWAG baseline results in the General Response. We hope it resolves your concerns and appreciate your constructive feedback.
>
> > Are the LLAMA results presented on MMLU 5-shot or 0-shot?
>
> The MMLU results pertain to the 0-shot case, and we realized this detail is only mentioned in the caption of Figure 7. We will update Appendix A to include this information. Thanks for bringing it to our attention!
>
> > It would also be great to see some additional analysis on the actual compute savings incurred using LPE-BSR.
>
> As advised, we have expanded the memory budgets plot from Section 4.4 to cover the experiments in Sections 4.2 and 4.5. Please refer to the figures in the attached PDF, which clearly illustrate the compute savings achieved with LPE-BSR, for further details. Thank you for the suggestion!

---

> > ### Comment · Reviewer_9d5B · 2024-08-12
> > **Thanks for the response**
> >
> > I thank the authors for their clarifications and for fixing the issues with SWAG results. I also find the new memory budget plots useful in actually conveying the tradeoffs of using the proposed approach versus model soups, and finetuning. I am going to keep my positive score.

---

> > > ### Author Response · Authors · 2024-08-12
> > >
> > > We are pleased to know that you found our additional clarifications and results helpful. We will definitely address all reviewer concerns in the final manuscript. Thank you once again for your constructive and supportive feedback!
> > >
> > > Sincerely,
> > > The authors of Submission 19991

---

### Author Rebuttal · Authors · 2024-08-06

# Global Response

First and foremost, we would like to thank all the reviewers for their time and effort in reviewing our paper. We are pleased to note that all the reviewers agreed our paper is of high quality. In particular, they noted that it tackles an important problem (yx79), is well-written and easy to follow (9d5B, yx79), has a generally good presentation (bKfp), is well-organized and clearly written (svQu), and includes extensive experiments and analyses (svQu). We are also delighted that our idea and methodology received favorable feedback, being described as inspired (9d5B), interesting (9d5B, bKfp), technically sound and effective (yx79).

While we will provide individual responses to each reviewer’s comments, this global response aims to address the major concerns raised by all reviewers. Due to the NeurIPS policy, we are unable to revise the paper or supplementary materials during the author response period. However, we assure that all points addressed during this period will be included in the camera-ready version.

---

## Further comparisons with other training-free baselines (bKfp, yx79)

Reviewers bKfp and yx79 emphasized the need for additional baselines. We acknowledge that incorporating Gaussian noise and Monte Carlo Dropout, as mentioned by the reviewers, can also be used for training-free ensemble construction. Accordingly, we have conducted an additional comparative study with CLIP-ViT-L/14, building upon the results presented in Table 3.

| Method      | NLL / ERR / ECE    | $\Delta$           |
| :-          | :-                 | :-                 |
| Pre-trained | .948 / .251 / .049 | -                  |
| MCDropout   | .938 / .251 / .041 | .010 / .000 / .008 |
| Gaussian    | .934 / __.250__ / .031 | .014 / __.001__ / .018 |
| LPE-BSR     | __.929__ / __.250__ / __.028__ | __.019__ / __.001__ / __.021__ |

The experimental results show that while both Gaussian and MCDropout baselines can also perform ensembling in a training-free manner, LPE-BSR achieves better performance. It is worth noting that LPE-BSR is more memory-efficient since each of its ensemble members uses INT-5, compared to FP32 used by the baseline methods. Therefore, LPE-BSR not only achieves better performance but also does so with reduced memory usage. We believe that updating the camera-ready version with these comparative results will enhance the discussion on training-free ensemble construction.

## Improving SWAG results (9d5B, svQu)

Reviewers 9d5B and svQu expressed concerns about the SWAG-Diag baseline results. After conducting additional experiments, we achieved improved baseline results for SWAG. Firstly, we found that introducing a scale hyperparameter to the variance matrix enhanced the SWAG-Diag results (cf. Appendix D.3 of the SWAG paper). The initially observed small variance matrix was likely due to our use of a small learning rate with the SGD optimizer without momentum. Secondly, we obtained SWAG results using a non-diagonal covariance matrix with a rank of ten. As Reviewer 9d5B pointed out, SWAG indeed outperforms SWAG-Diag.

| Method               | NLL / ERR / ECE    | $\Delta$           |
| :-                   | :-                 | :-                 |
| MAP (SWA)            | .488 / .137 / .034 | -                  |
| SWAG-Diag (previous) | .487 / .137 / .034 | .001 / .000 / .000 |
| SWAG-Diag (improved) | .479 / __.136__ / .027 | .009 / __.001__ / .007 |
| SWAG (rank=10)       | .477 / __.136__ / __.021__ | .011 / __.001__ / __.013__ |
| LPE-BSR              | __.473__ / __.136__ / __.021__ | __.015__ / __.001__ / __.013__ |

However, LPE-BSR achieves performance on par with SWAG while being more memory-efficient, emphasizing the potential of low precision ensembling. We believe these improved baseline results further underscore our contribution, showing that LPE-BSR generates low precision ensemble members that are comparable to the posterior samples from leading Bayesian methods like SWAG and IVON.

## Updated figures

We are providing the updated figures as a PDF in line with NeurIPS policy. Specifically:
- Figure 9 is a revised version of the original Figure 2, featuring an improved SWAG baseline and including the previously missing error bars for BMA.
- Figure 10 shows the memory budgets for ensembling compared to the MAP checkpoint and BMA baselines.
- Figure 11 also presents the memory budgets for ensembling in relation to the pre-trained checkpoint. We have corrected the pre-trained results for CLIP-ViT-G/14 from .948 to .942; Figure 7 will be updated accordingly. LPE-BSR effectively lowers the memory budgets compared to baselines.
- Figure 12 builds on Figure 5 by introducing more advanced ensembling methods rather than fast ensembling techniques like SSE. DE represents an ensemble of multiple Adam solutions, while MultiIVON represents an ensemble of multiple IVON solutions. LPE-BSR consistently improves upon these ensemble methods.

---

> ### Comment · Reviewer_bKfp · 2024-08-12
> **details about experiments with MCDropout and Gaussian noise**
>
> Thank you for your additional experiments. Could you please provide more details about the hyper-parameters of MCDropout and Gaussian noise used to build the ensemble?
> Which hyper-parameters did you consider? Which values?
> Did you select the optimal hyperparameters? Did you put as much effort in finding the optimal hyper-parameters as in your method?
> Thanks

---

> > ### Author Response · Authors · 2024-08-12
> >
> > We appreciate Reviewer bKfp's additional question about the hyperparameters of MCDropout and Gaussian baselines and are happy to provide more detailed results:
> >
> > | Method               | NLL / ERR / ECE    |
> > | :-                   | :-                 |
> > | Pre-trained          | .948 / .251 / .049 |
> > ||
> > | LPE-BSR (INT-5)      | __.929__ / __.250__ / __.028__ |
> > | MCDropout (p=0.001)  | .946 / .251 / .047 |
> > | MCDropout (p=0.002)  | .946 / .251 / .047 |
> > | MCDropout (p=0.004)  | .944 / __.250__ / .046 |
> > | MCDropout (p=0.008)  | .940 / __.250__ / .044 |
> > | __MCDropout (p=0.016)__  | __.938__ / .251 / .041 |
> > | MCDropout (p=0.032)  | .944 / .255 / __.034__ |
> > ||
> > | LPE-BSR (INT-5)      | __.929__ / __.250__ / __.028__ |
> > | Gaussian (σ^2=0.0001)  | .948 / .251 / .049 |
> > | Gaussian (σ^2=0.0002)  | .948 / .251 / .048 |
> > | Gaussian (σ^2=0.0004)  | .946 / __.250__ / .046 |
> > | Gaussian (σ^2=0.0008)  | .941 / __.250__ / .043 |
> > | __Gaussian (σ^2=0.0016)__  | __.934__ / __.250__ / __.031__ |
> > | Gaussian (σ^2=0.0032)  | .981 / .264 / .038 |
> >
> > Here, p refers to the drop probability in MCDropout, and σ^2 denotes the variance of Gaussian noise. The previously reported values were obtained by identifying p or σ^2 values that are optimal--not too small or too large--where the ensemble benefit is greatest. We assure to further clarify this point and include it in the final manuscript.
> >
> > Sincerely,
> > The authors of Submission 19991

---

### Decision · Program_Chairs · 2024-09-25

**Decision:**

Accept (poster)

**Comment:**

As stated by the Authors, the main contribution of the paper is offering a fresh perspective by using quantization errors--often seen as challenges in quantization research--as a source of diversity in ensemble methods.

This approach has a particular significance in the era of LLMs. Ensembling is known to improve robustness and the proposed approach makes ensembling more accessible for LLMs due to the reduce memory requirement.

The paper attracted diverging reviews, with two reviewers voting for acceptance (6,5) and two voting for rejection (4,4). It is a borderline paper and the decision will naturally be based more on the highlighted weaknesses and strengths, and less on the absolute score.

Two reviewers emphasized the lack of novelty, which I respectfully disagree with. If the previous work does not disclose the proposed method, the fact it is simple is actually more of a strength of the work rather than a limitation. I would also argue that the results suggest that quantization of the model has interesting and nontrivial effects on its robusstness. It would be interesting to study how these different ensemble members differ, e.g. in terms of which examples they memorize from the original dataset.

A very fair criticism was raised that the paper lacked comparisons to methods such as SWAG, Deep Ensemble or Batch Ensemble. This I think was convincingly addressed in the rebuttal phase.

All in all, I am recommending acceptance. I believe the method might have an impact, both for applied deep learning and development of new methods.